# ExDBN: Learning Dynamic Bayesian Networks using Extended Mixed-Integer Programming Formulations

**Pavel Rytíř**  *rytir.pavel@fel.cvut.cz*
*Czech Technical University in Prague*

**Aleš Wodecki**  *wodecki.ales@fel.cvut.cz*
*Czech Technical University in Prague*

**Georgios Korpas**  *georgios.korpas@hsbc.com.sg*
*HSBC Holdings Plc., Singapore*
*Czech Technical University in Prague*
*Archimedes AI, Athena Research Center, Greece*

**Jakub Mareček**  *jakub.marecek@fel.cvut.cz*
*Czech Technical University in Prague*

**Reviewed on OpenReview:** *https://openreview.net/forum?id=I64MJzl9Fy*

## Abstract

Causal learning from data has received much attention recently. Bayesian networks can be used to capture causal relationships. There, one recovers a weighted directed acyclic graph in which random variables are represented by vertices, and the weights associated with each edge represent the strengths of the causal relationships between them. This concept is extended to capture dynamic effects by introducing a dependency on past data, which may be captured by the structural equation model. This formalism is utilized in the present contribution to propose a score-based learning algorithm. A mixed-integer quadratic program is formulated and an algorithmic solution proposed, in which the pre-generation of exponentially many acyclicity constraints is avoided by utilizing the so-called branch-and-cut ("lazy constraint") method. Comparing the novel approach to the state-of-the-art, we show that the proposed approach turns out to produce more accurate results when applied to small and medium-sized synthetic instances containing up to 80 time series. Lastly, two interesting applications in bioscience and finance, to which the method is directly applied, further stress the importance of developing highly accurate, globally convergent solvers that can handle instances of modest size.

## 1 Introduction

Dynamic Bayesian Networks (DBNs) were conceived by Dagum et al. (1992) to unify and extend traditional linear state-space models, linear and normal forecasting models (such as ARMA), and simple dependency models (such as hidden Markov models) into a general probabilistic representation and inference mechanism for arbitrary nonlinear and non-normal time-dependent domains. Murphy (2002) made important contributions on the inference. The network may be used to compute posterior probabilities for different situations, or the learned graph structure may be inspected, and dependencies of particular interest analyzed. This explainability is a key benefit of DBN. To the present day, there are no scalable methods for learning DBNs.

There exist a number of methods for learning DBN. One can learn underlying continuous dynamics represented by stochastic differential equations (Bellot et al., 2021), or related dynamical systems (Murphy, 2002). Another option is to assume a priori knowledge about time-lagged data and incorporate this knowledge into the solver (Sun et al., 2021). Furthermore, one deal with the general problem and propose local

methods (Pamfil et al., 2020; Gao et al., 2022), which can scale further at the cost of some loss of accuracy. Under the assumptions of Markovianity and faithfulness, conditional independence-based methods, such as PCMCI+ (Runge, 2020), can be applied. Note that many of the previous works also combine several of these approaches to design solvers that are efficient and applicable to a wide range of applications. However, it should be noted that many methods may not identify DAG representations of causal dependencies under certain conditions (Kaiser & Sipos, 2022; Reisach et al., 2021). One of the possible causes is that many of them only converge to a first-order stationary point for the corresponding non-convex optimization problem, which is known (Daniely & Shalev-Shwartz, 2016) to be hard to solve.

In our contribution, we revisit the score-based learning of DBNs utilizing a directed acyclic graph (DAG) structure augmented by additional time lagged dependencies (Murphy, 2002; Dean & Kanazawa, 1989; Assaad et al., 2022). We utilize mixed-integer programming to learn the underlying dynamic Bayesian networks. While all of the previous methods focus mostly on scaling with adequate precision by utilizing a variety of heuristics, we focus on leveraging quadratic mixed-integer programs to find near global optimum solutions to a score-based DAG learning problem, which results in a high-quality reconstruction of the DAG.

Notice that the number of constraints needed enforce the structure of a directed acyclic graph ("cycle-exclusion constraints") is super-exponential in the number of time series on the input. This is sometimes referred to as the curse of dimensionality. We tackle the curse of dimensionality by avoiding the pre-generation of these cycle-exclusion constraints and subsequently adding these constraints only if needed. It is shown that, only a small number of these constraints are actually needed to ensure the acyclicity of the directed graph, which allows for a large speedup over the naive version of the algorithm that pre-generates all of the constraints. Additionally, this technique allows the method to solve problems much larger compared to the naive counterpart, since the memory consumption is greatly reduced. The formulation and its implementation are easily reproducible, making it accessible to a wide range of potential practitioners.

Learning DBNs has been successfully applied to a variety of problems, many of which are related to applications in medicine (Zandonà et al., 2019; van Gerven et al., 2008; Michoel & Zhang, 2023; Zhong et al., 2023). In addition to medical applications, dynamic Bayesian networks are widely used in econometrics (Demiralp & Hoover, 2003) and financial risk modeling (Ballester et al., 2023). The applications of DBN are not limited to the ones mentioned and the curious reader may refer to (Kungurtsev et al., 2024) for further details. After we present our approach, we revisit these applications in Sections 4.5 and 4.4.

## 2 Problem Formulation

Before formulating the problem of score-based Bayesian network learning as a mixed-integer program, let us describe the problem as a structural vector autoregressive model (Demiralp & Hoover, 2003; Kilian, 2011). Let $d \in \mathbb{N}$ be the number of variables in an autoregressive model. Let $T \in \mathbb{N}$ be the total number of time periods and let $X_{i,t}$ be a set of endogenous variables at the given time $t$, where $i \in \{1, 2, \ldots, d\}$ and $t \in \{1, 2, \ldots, T\}$. We assume that the variables interact with each other both simultaneously and with time lags. These interactions can be captured by the following equation:

$$X_t = X_t W + X_{t-1} A_1 + X_{t-2} A_2 + \ldots + X_{t-p} A_p + Z_t, \tag{1}$$

where $p \in \mathbb{N}$ is the autoregressive order, and

$$W \in \mathbb{R}^{d \times d}, A_i \in \mathbb{R}^{d \times d}, Z_t \in \mathbb{R}^d, i \in \{1, 2, \ldots, p\}.$$

Simultaneous interactions are called intra-slice and are represented by the matrix $W$ and time-lagged interactions are called inter-slice and are represented by the matrices $A_i$, where $i$ is the number of timesteps (length) of the time lag. The vector $Z_t$ is an error term. We will not assume any particular distribution of error terms.

We will assume that there is no cycle in intra-slice interactions. In other words, if we view $W$ as an adjacency matrix of a directed graph $G$, then $G$ is a directed acyclic graph (DAG). The matrices $A_i$ could also be viewed as adjacency matrices of directed graphs, but we do not make assumptions about these graphs. Note that non-linear autoregressive models can also be formulated in an analogous way.

Our goal is to learn the matrices $W$, which corresponds to a DAG, and $A_i$ for $i = 1, \ldots, p$. Assume that we have $n$ data samples of each of $d$ random variables organized into a data matrix $X \in \mathbb{R}^{n \times d}$. Then, the following equation holds for the solution $W, A_i, i = 1, \ldots, p$.

$$X = XW + Y_1 A_1 + \cdots + Y_p A_p + Z, \tag{2}$$

where $Y_i$ are time lagged version of $X$ for $i = 1, \ldots, p$ and $Z \in \mathbb{R}^{n \times d}$ is a matrix with error terms for all samples.

To maximize the fit of the data over the model, we want to minimize the Frobenius norm of the error matrix $Z$. We formulate it as the following cost function:

$$J(W, A_1, \ldots, A_p) \quad = \quad \|X - XW - Y_1 A_1 - \cdots - Y_p A_p\|_F^2 \;+\; \lambda \|W\| \;+\; \eta \left(\|A_1\| + \cdots + \|A_p\|\right), \tag{3}$$

where $\|\cdot\|_F$ denotes Frobenious matrix norm and $\|\cdot\|$ denotes an arbitrary matrix norm and $\lambda, \eta > 0$ are sufficiently small regularization coefficients. The data fitting problem is as follows

$$
\begin{aligned}
\min_{W, A_1, \ldots A_p} \quad & J(W, A_1, \ldots, A_p) \\
& W \text{ is acyclic}, W \in \mathbb{R}^{d \times d}, \\
& A_i \in \mathbb{R}^{d \times d}, i \in \{1, 2, \ldots, p\}.
\end{aligned}
\tag{4}
$$

**Remark 1** *The identifiability of $W$ and $A_i$ using (4) has been studied for Gaussian and non-Gaussian noise. Regardless of noise, the identifiability of $A$ is a consequence of the basic theory of autoregressive models (Kilian, 2011). The identifiability of $W$ is a bit more involved and must be separated into the Gaussian and non-Gaussian case. However, in either case, identifiability is possible under mild conditions (Hyvärinen et al., 2010; Peters & Bühlmann, 2012).*

Note that if the autoregressive order p is set to 0, ExDBN could be used to learn (static) Bayesian Networks. Dynamic Bayesian Networks differ from Bayesian Networks (BNs) primarily in the statistical structure of their data. While BNs are trained on sets of independent and identically distributed samples, DBNs rely on trajectories of time series where successive samples may be temporally dependent. The inference goal of DBN models is typically forecasting, whereas for BNs it is generative modeling. To evaluate learning algorithms, different types of benchmarks are required for BNs and DBNs. Therefore, in this work, we focus strictly on DBNs, i.e., autoregressive orders $p \geq 1$.

## 2.1 Brief Introduction To Mixed Integer Quadratic Programming

To better frame the content of Section 2.2, we provide a short introduction to mixed-integer quadratic programming. An optimization problem, is called a mixed-integer quadratically constrained quadratic program (MIQCQP) if it is of the form

$$\min_{x \in \mathbb{R}^n} \quad x^T Q x + q^T x, \tag{5}$$

$$\text{s.t.} \quad x^T Q_i x + q_i^T x \leq a_i, \tag{6}$$

$$Ax \leq b, \tag{7}$$

$$x_i \in \mathbb{R} \text{ for } i \notin I \tag{8}$$

$$x_i \in \mathbb{Z} \text{ for } i \in I \tag{9}$$

where $Q, Q_i \in \mathbb{R}^{n \times n}$, $q, q_i \in \mathbb{R}^n$, $A \in \mathbb{R}^{m \times n}$, $a \in \mathbb{R}^k$, $b \in \mathbb{R}^m$, for some $m, n, k, r \in \mathbb{N}$. Expression (5) is often called the objective, cost, or loss function, Inequality (6) represents the quadratic constraints, Inequality (7) are the linear constraints, and index set $I$ denotes the integral variables.

Mixed-integer quadratic programs have been shown to be in NP (Del Pia et al., 2014), which often leads to an exhaustive demand for computational resources. The algorithms used to solve MIQP are typically

branch-and-bound or cutting plane (Dakin, 1965; Bonami et al., 2009; Westerlund & Pettersson, 1995; Kronqvist et al., 2015). Both of these algorithmic treatments are often employed together, often with the addition of a presolving step, the use of heuristics and parallelism. The aforementioned allows many modern solvers to solve even large problems despite the NP hardness. Some of these solvers are open source (like SCIP (Achterberg, 2009) and GLPK (GNU Project, 2025)) and others are commercial (GUROBI (Gurobi Optimization, LLC, 2024) and CPLEX (IBM, 2025)). The powerful infrastructure present in these solvers can be made use of together with additional problem-specific modifications to deliver high-quality solutions.

Due to the exhaustive nature of the algorithms mentioned in the previous paragraph, global convergence is guaranteed (Belotti et al., 2013). Furthermore, convergence to the global solution may be tracked and the error estimated by computing the dual problem of (5)–(9). The dual of the problem is then used to computed the so called MIP gap as follows

$$\text{MIP gap} = \frac{|J(x^*) - J_{\text{dual}}(y^*)|}{|J(x^*)|},\tag{10}$$

where $x^*$ and $y^*$ are the current best solutions of the primal and dual problems respectively, and $J$ and $J^*$ are the cost functions of the primal and dual problems, respectively. The MIP GAP ensures that we can assess the quality of the minimization during solution time and terminate the computation when the result is good enough (small enough MIP GAP). Furthermore, if the gap reaches 0 at any point, we are sure that the current solution is a global optimum.

## 2.2 Mixed Integer Quadratic Programming Formulation

Formulating the learning problem allows us to use a globally convergent algorithm. Key formulations for the learning of directed acyclic graphs, a closely related problem, have been proposed by Manzour et al. (2021); Xu et al. (2024); Jaakkola et al. (2010).

First, we define the program variables. For $i, j = 1, \ldots, d$, we define the binary variable $e_{ij}$ whose value is interpreted as follows: $e_{ij}$ equals 1 if and only if there is an oriented edge from node $i$ to node $j$ both in the intra-slice graph.

Similarly, we define binary variables for edges connecting nodes from inter-slices graphs to the nodes of intra-slice graph. For $i, j = 1, \ldots, d$ and $t = 1, \ldots, p$, where $p$ is the autoregressive order. We define the binary variable $e_{ij}^t$ whose value is interpreted as follows: $e_{ij}^t$ equals 1 if and only if there is an oriented edge from node $i$ in the inter-slice graph, corresponding to time lag $t$, to the node $j$ in the intra-slice graph.

With each of the above variable $e_{ij}$ and $e_{ij}^t$, we associate a continuous variables $w_{ij}$ and $a_{ij}^t$, respectively, which encodes the weight of the corresponding edge.

Using these variables, the scoring function of problem (4) becomes the following MIQP objective:

$$J(W, A_1, \ldots, A_p) = \sum_{i=1}^n \sum_{j=1}^d \left( X_{ij} - \sum_{k=1}^d X_{ik} w_{kj} - \sum_{t=1}^p \sum_{k=1}^d Y_{t,ik} a_{kj}^t \right)^2 + \text{REG},\tag{11}$$

where REG is the regularization part of the objective function. We use two different regularization functions L0 and L2 defined as follows. Regularization L0 is defined as:

$$\text{REG} = \lambda \sum_{i=1}^n \sum_{j=1}^n e_{i,j} + \eta \sum_{s=1}^p \sum_{i=1}^n \sum_{j=1}^n e_{i,j}^s.\tag{12}$$

Regularization L2 is defined as:

$$\text{REG} = \lambda \sum_{i=1}^n \sum_{j=1}^n (w_{i,j})^2 + \eta \sum_{s=1}^p \sum_{i=1}^n \sum_{j=1}^n (a_{i,j}^s)^2,\tag{13}$$

where $\lambda > 0$ and $\eta > 0$ are regularization coefficients.

Next, we define the constraints of the optimization problem. In order to ensure that $W$ encodes a directed acyclic intra-slice graph. We add the following constraints (Dantzig et al., 1954) that ensure that $W$ is acyclic.

Let $\mathcal{C}$ denote the set of all directed cycles in the complete directed graph on $d$ vertices without loops. Then we add the following constraints:

$$\sum_{(i,j)\in C} e_{i,j} \leq |C| - 1 \text{ for all } C \in \mathcal{C} \tag{14}$$

Note that, the number of above constraints is exponential. We will discuss in the following section how to deal with such number of constraints.

In order to relate the binary variables $e_{ij}$ and $e_{ij}^t$ with the weight variables $w_{ij}$ and $a_{ij}^t$, we introduce the following constraints:

$$\begin{aligned} w_{ij} &\leq ce_{ij}, \\ w_{ij} &\geq -ce_{ij} \quad \text{for all } i,j \in \{1,2,\ldots,d\}, \end{aligned} \tag{15}$$

$$\begin{aligned} a_{ij}^t &\leq ce_{ij}^s \quad \text{for all } k,j \in \{1,2,\ldots,d\}, \\ a_{ij}^t &\geq -ce_{ij}^s \quad \text{for all } t \in \{1,2,\ldots,p\}, \end{aligned} \tag{16}$$

where $c > 0$ is the maximal admissible magnitude of any weight. In our numerical experiments, we choose $c$ large enough to not affect the result of the learning.

## 3 Algorithmic Implementation Using Branch-and-Bound-and-Cut

One of our main contributions is the development of a branch-and-bound-and-cut algorithm to solve the formulation mentioned above. The Branch-and-Bound-and-Cut algorithm (Achterberg, 2007, e.g.) solves mixed-integer optimization problems by combining three key steps: solving linear relaxations to obtain bounds, branching on variables when solutions are fractional, and tightening relaxations with valid inequalities. In addition, users can supply so-called "lazy" constraints, which are constraints not included in the initial model but enforced only when violated by a candidate integer solution during the search. This lets the solver keep the model smaller and only add these constraints when necessary, further improving efficiency while guaranteeing correctness.

Since the acyclic constraints (14) need to be imposed only for the edges of the graph representing the intra-slice level, all of what follows is only applied to the intra-slice graph. While we leverage the traditional branch-and-bound approach as described in (Achterberg, 2007, e.g.), we incorporate cycle exclusion constraints (14) using "lazy" constraints. These are only enforced once an integer-feasible solution candidate is found. If a violation of a lazy constraint occurs, the constraint is added across all nodes in the branch-and-bound tree. At the root node, only $O(|E|)$ constraints (15) and (16) are initially used. Cycle-exclusion constraints (14) are added later. Note that this method is not a heuristic and does not lead to a possibly harmful reduction (or extension) of the solution space leading to omitting possible solutions or returning solutions which are not DAGs. Furthermore, it is shown that the number of constraints that are actually needed in a computation is many orders of magnitude less than the number of all possible constraints.

Once a new mixed-integer feasible solution candidate is identified, detecting cycles becomes straightforward using a depth-first search (Even & Even, 2011, pp. 46–48). If a cycle is detected, the corresponding lazy constraint (14) is added to the problem. The depth-first search (DFS) algorithm solves the problem of cycle detection in a worst-case quadratic runtime relative to the number of vertices in the graph, which contrasts with algorithms that separate related inequalities from a continuous relaxation (Borndörfer et al., 2020; Cook et al., 2011), such as the quadratic program in our case. Three variants of adding lazy constraints for the problem were tested.

- Adding a lazy constraint only for the first cycle found.

- Adding a lazy constraint only for the shortest cycle found.

- Adding multiple lazy constraints for all cycles found in the current iteration in which an integer-feasible solution candidate is available.

The third mentioned variant was found to consistently deliver the best results, despite (Achterberg, 2007, Chapter 8.9). See Table 2 for numerical results. Therefore, it is applied in all the numerical tests that follow.

## 4 Numerical Experiments

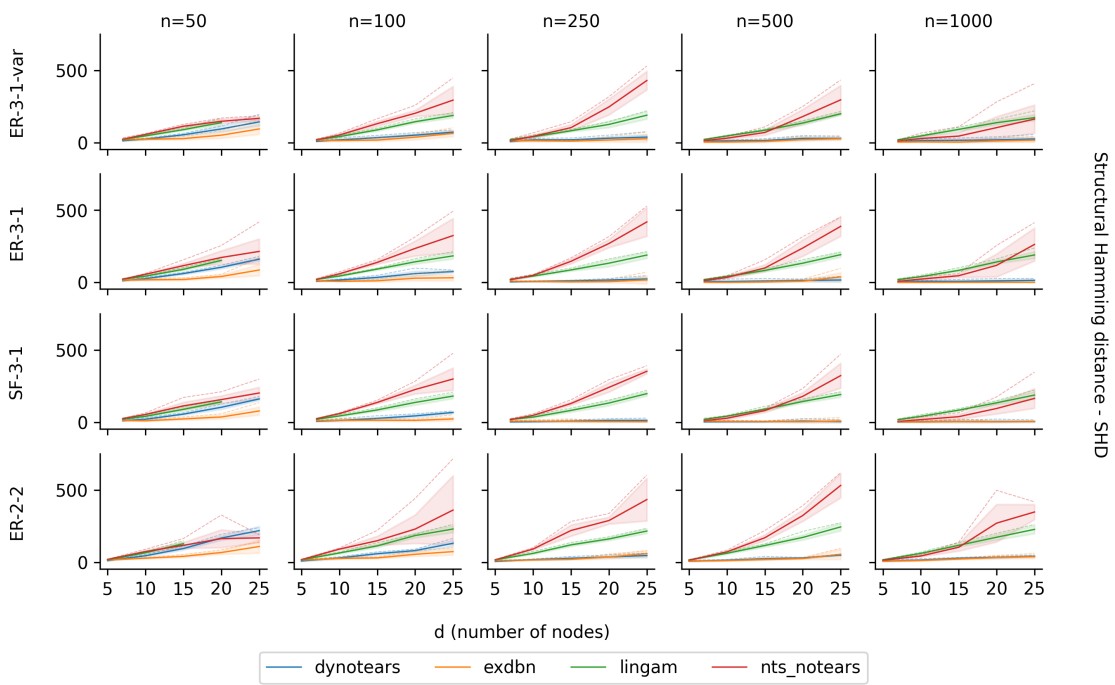

Figure 1: SHD for a test cases using ER-3-1, SF-3-1, ER-2-2 random ensembles, where the first number is the edge-vertex ratio on intra graph, the second is the autoregressive order. The edge-vertex ratio of inter graph is always 1. For the problem denoted by var suffix, the variance of Gaussian noise is randomly sampled from uniform distribution on interval $(0.6, 1.2)$ for each variable. For the other problems, the variance is set to 1. The mean, standard deviation, and maximum over 10 algorithm runs is depicted. Each run is performed on different randomly sampled dataset.

In recent years, many solvers have been developed to facilitate the graphical learning of Bayesian networks that represent causality (Pamfil et al., 2020; Hyvärinen et al., 2010; Malinsky & Spirtes, 2018; Gao et al., 2022; Dallakyan, 2023; Lorch et al., 2021). Each of these solvers (including the one presented) faces the curse of dimensionality, which somewhat restricts their applicability; thus, thorough testing is required. It is impossible to test the proposed solution against every solver that has been developed. However, there is a significant branch of research that allows for direct comparison, and by the transitivity of results, this enables indirect comparison with many previous solvers.

Pamfil et al. (2020) have developed a locally convergent method, called DYNOTEARS, that learns causality as a Bayesian network that supersedes the solution methods previously developed (Hyvärinen et al., 2010; Malinsky & Spirtes, 2018; Zheng et al., 2018). Further developments based on previous publications include formulating the problem in the frequency domain or defining differentiable Bayesian structures (Dallakyan, 2023; Lorch et al., 2021). In this section, we provide a comparison with DYNOTEARS, LiNGAM (Hyvärinen

et al., 2010), and NTS-NOTEARS (Sun et al., 2021), and thus, by transitivity, with the methods documented by Hyvärinen et al. (2010); Malinsky & Spirtes (2018).

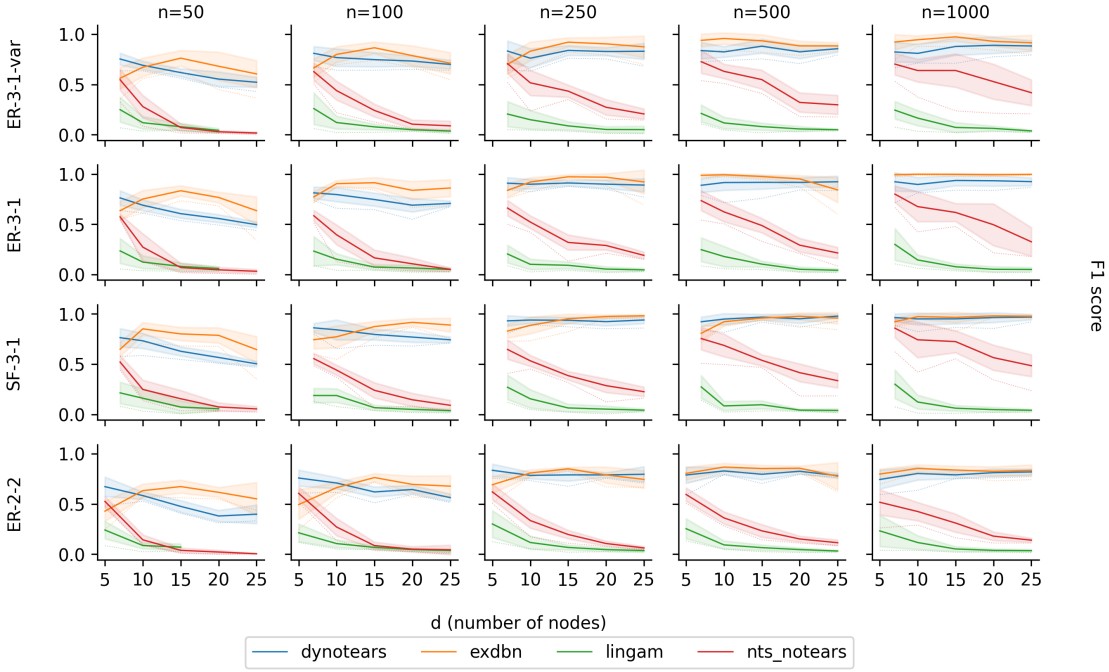

Figure 2: F1 score for test cases using ER-3-1, SF-3-1, ER-2-2 random ensembles.

## 4.1 Synthetic Data

One of the evaluations of ExDBN was performed on the synthetic data generated by the following process. First, a random intra-slice directed acyclic graph (DAG) was generated using either the Erdős-Rény (ER) model or the scale-free Barabási–Albert (SF) model. Then, the DAG weights were sampled uniformly from the union of intervals $[-2.0, -0.5] \cup [0.5, 2.0]$.

Next, the inter-slice graphs were generated using the ER model. For each inter-slice graph, weights were sampled from the interval $[-0.5\alpha, -0.2\alpha] \cup [0.2\alpha, 0.5\alpha]$, where $\alpha = 1/\eta^{t-1}$, $\eta \geq 1$ is the decay parameter, and $t$ is the time lag of the slice. $t = 0$ corresponds to the intra-slice, while $t \in \{1, \ldots, p\}$ represents the inter-slices.

The data samples are then generated using the structural equation model equation (1) and adding Gaussian noise with either variance 1 or different variance for each variable sampled uniformly from a given interval.

## 4.2 Benchmark Setup and Quantities of Interest

Let $W_{\text{true}}$ denotes the adjacency matrix representing the intra-slice ground truth graph of a given problem and let $A_{t,\text{true}}$ denotes the adjacency matrices of inter-slice ground truth graphs of time-lagged interactions. In the case of synthetic data, $W_{\text{true}}$ and $A_{t,\text{true}}$ are known, since they were used for data generation.

Let $W_{\text{est}}$ and $A_{t,\text{est}}$ be the solution of the learning problem. We will evaluate their quality by comparing to the ground truth and computing structural Hamming distance and $F_1$ score.

We apply a small threshold $\epsilon = 0.15$ to $W_{\text{est}}$ and $A_{t,\text{est}}$. We set all the elements that are smaller than $\epsilon$ to 0. We do that in order to remove small rounding errors that would negatively affect computation of discrete metrics such as structural Hamming distance.

Structural Hamming distance (SHD) is defined as follows:

$$\text{SHD}\left(W_{\text{true}}, A_{t,\text{true}}; W_{\text{est}}, A_{t,\text{est}}\right) = \sum_{i,j=1}^{d} \text{dist}_{ij}\left(W_{\text{true}}, W_{\text{est}}\right) + \sum_{t=1}^{p}\sum_{i,j=1}^{d} \text{dist}_{ij}\left(A_{t,\text{true}}, A_{t,\text{est}}\right), \quad (17)$$

where

$$\text{dist}_{ij}\left(C, D\right) = \begin{cases} 0 & \text{if } C_{ij} \neq 0 \text{ and } D_{ij} \neq 0 \\ 0 & \text{if } C_{ij} = 0 \text{ and } D_{ij} = 0 \\ \frac{1}{2} & \text{if } C_{ij} \neq 0 \text{ and } D_{ji} \neq 0 \\ 1 & \text{otherwise.} \end{cases} \quad (18)$$

SHD is used as a score that describes the structural similarity of two DAGs in terms of edge placement and is commonly used to assess the quality of solutions (Zheng et al., 2018; Pamfil et al., 2020). Besides SHD,

$$\text{precision} = \frac{\text{true positive}}{\text{true positive} + \text{false positive}} \quad (19)$$

$$\text{recall} = \frac{\text{true positive}}{\text{true positive} + \text{false negative}}, \quad (20)$$

are used (Andrews et al., 2024) to evaluate the quality of structural recovery. It is important to note that precision and recall isolate false positives and negatives, respectively, in contrast to SHD, where these quantities are both accounted for simultaneously. The last metric that can be used to evaluate structural similarity is the $F_1$ score and reads

$$F_1 = \frac{2}{\text{precision}^{-1} + \text{recall}^{-1}}. \quad (21)$$

Note that all of the quantities evaluated in (19) and (21) are a result of summing up all of the differences over both inter-slice and intra-slice matrices between the solution and the ground truth.

### 4.3 Synthetic Benchmark Results

In the following benchmark, the generation methods described in Section 4.1 are used to compare ExDBN with DYNOTEARS, LiNGAM, and NTS-NOTEARS under the assumption of Gaussian noise. Even though the cost function is a maximum likelihood estimator (see Section 1) for non-Gaussian noise, we leave this evaluation for future publication. The performance was studied for different numbers of variables, samples, and graph generation methods, with the relevant metrics: SHD and F1 score, reported in Figures 1 and 2.

A statistical ensemble with 10 different seeds was used for each experiment, and the mean, standard deviation, and worst-case values are reported in the plots. We used Gurobi as MIQP solver. The running time for

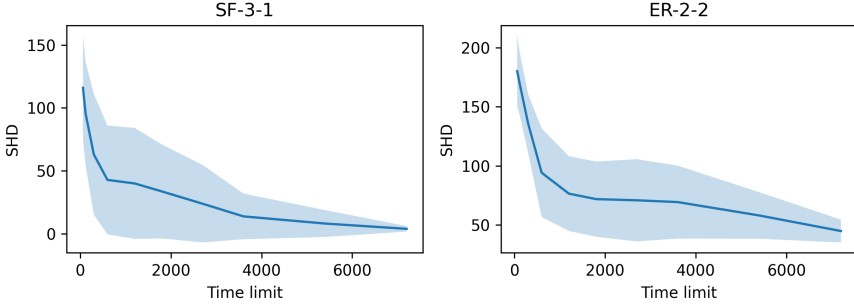

Figure 3: Comparison of ExDBN solution quality (SHD) and running time for the ER-2-2 and SF-3-1 ensembles with 25 variables and 250 samples. The mean, and standard deviation over 10 algorithm runs is depicted. Each run is performed on different randomly sampled dataset.

| Number of nodes | DYNOTEARS | ExDBN | LiNGAM | NTS-NOTEARS |
|---|---|---|---|---|
| 5 | $1.1 \pm 0.4$ | $530.5 \pm 1504.2$ | $0.7 \pm 0.4$ | $74.7 \pm 52.6$ |
| 7 | $2.0 \pm 0.8$ | $2188.7 \pm 3115.6$ | $0.6 \pm 0.2$ | $142.5 \pm 87.4$ |
| 10 | $4.5 \pm 2.0$ | $5163.1 \pm 3014.5$ | $0.7 \pm 0.2$ | $379.9 \pm 237.8$ |
| 15 | $12.2 \pm 5.1$ | $7214.6 \pm 66.4$ | $1.0 \pm 0.2$ | $875.6 \pm 490.1$ |
| 20 | $24.9 \pm 11.8$ | $7216.1 \pm 182.0$ | $1.7 \pm 0.3$ | $1571.4 \pm 953.4$ |
| 25 | $47.0 \pm 27.1$ | $7226.3 \pm 20.7$ | $2.9 \pm 0.6$ | $1975.7 \pm 1114.5$ |

Table 1: Running times of the evaluated methods (mean $\pm$ standard deviation) across all scenarios, sample sizes, and ten runs on different randomly generated datasets.

ExDBN was capped at 7200 seconds, and the memory was limited to 32 GB. The regularization applied in ExDBN needs to be scaled appropriately with the number of samples, as the optimal regularization constant is assumed to be a decreasing function of sample size. We use this assumption as a non-strict guideline to select the appropriate regularization for a given sample size. This follows from the requirement that regularization should remain proportionally small compared to the main objective expressed by equation (11). It was also found that switching from L1 to L2 regularization improves identification performance when the number of samples is large. When the ground-truth graph is unknown, the algorithm can be run for multiple values of $\lambda$ and $\eta$, and the configuration yielding a better MIP GAP can be selected. For smaller sample sizes, L1 regularization performs better, whereas for larger sample sizes, L2 regularization typically yields good results and is faster.

As noted in (Reisach et al., 2021), the noise variances and data scale may be important for some algorithms to perform well. We tested ExDBN on normalized data and noticed a significant performance drop. Therefore, ExDBN is suitable for problems in which the data of the samples have a true scale.

The results of the tests can be divided into two categories based on the variance of the applied noise. In the non-equal variance scenario, ExDBN performs better than DYNOTEARS (see Figures 1 and 2). In the scenario where all variances are equal to 1, the results are more balanced between ExDBN and DYNOTEARS. LiNGAM and NTS-NOTEARS perform substantially worse than both ExDBN and DYNOTEARS. This can be attributed to the fact that LiNGAM is optimized for non-Gaussian data, while NTS-NOTEARS is designed for non-linear models. Note that ExDBN shows a higher standard deviation in some scenarios with 25 variables. This deviation may be reduced by allowing more computation time for ExDBN, as suggested by Figure 3, which compares standard deviation and running time.

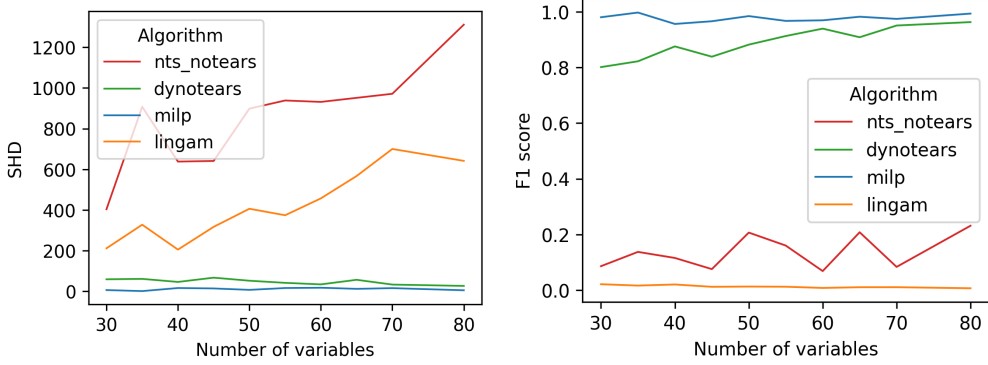

Figure 4: SHD and F1 score on the SF-3-1 benchmark for larger problem instances. The number of variables $d$ ranges from 30 to 80, and the number of samples $n$ is set to $5d$.

Focusing on the 50-sample case, while also considering the previous results, we observe that the performance gap between the solvers widens in favor of ExDBN as the number of variables increases. In the lower-sample

scenarios, ExDBN outperforms DYNOTEARS for many graph sizes on average and consistently outperforms DYNOTEARS in the worst-case results (minimum or maximum, depending on the metric).

Note that the global convergence of the method, which is rooted in the fundamentals of mixed-integer quadratic programming, allows for increased computation time, leading to further improvements in the reported metrics. While some time-sensitive applications, such as short-term stock evaluation, may not be able to benefit from this, others—such as biomedical applications—can, as computations lasting several days with measurably improved accuracy (monitored via the duality gap) are often desirable. See Figure 3 for a comparison of running time and solution quality, and Table 1 for a comparison of the running times of the evaluated methods.

To evaluate the scalability of ExDBN, we conducted experiments on a simplified benchmark based on SF-3-1. The number of variables $d$ ranged from 30 to 80, and the number of samples was set to $5d$. The running time was limited to 22 hours, and the memory limit was increased to 128 GB. See Figure 4 for the results. We can conclude that within these limits, ExDBN scales up to 80 variables and performs better than the other compared methods on this benchmark.

| Lazy constraints added for | mean MIP GAP |
|---|---|
| all cycles found | 0.335 |
| the shortest cycle found | 0.344 |
| the first cycle found | 0.346 |

Table 2: Evaluation of variants for adding lazy constraints on SF-3-1

We ran experiments on three variants of adding lazy constraints on SF-3-1. The results are shown in Table 2. They indicate that adding lazy constraints for all cycles found in the current iteration performs best on average. See Figure 5 for the number of added lazy constraints during the ExDBN run in each scenario.

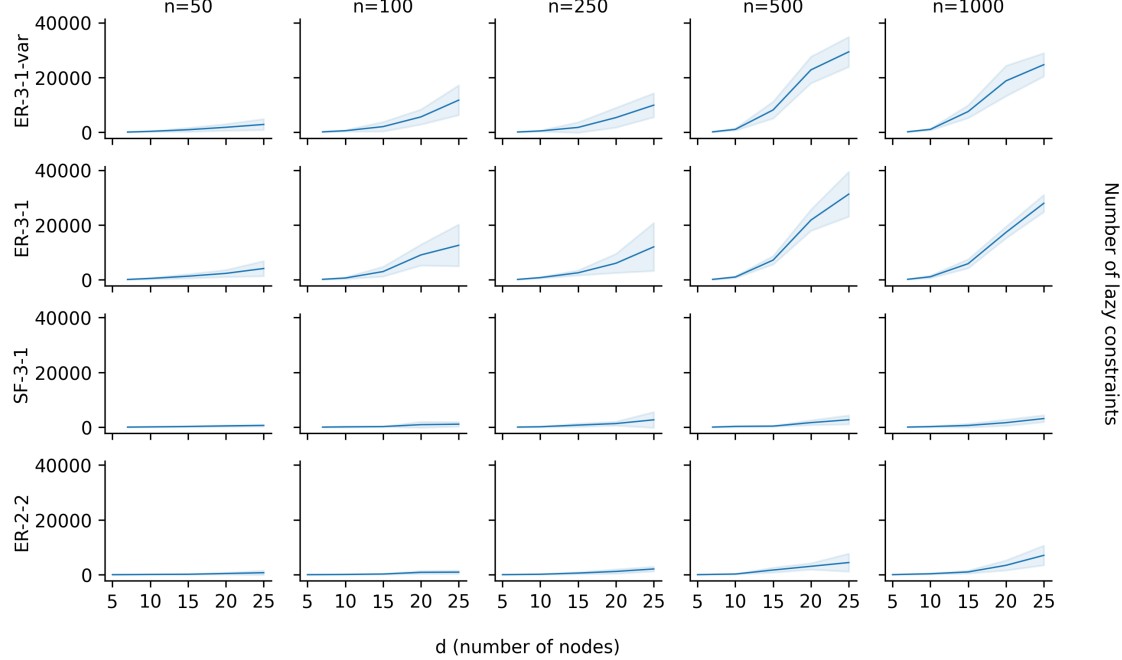

Figure 5: Number of lazy constraints added by ExDBN using "all cycles" variant on test cases using ER-3-1, SF-3-1, ER-2-2 random ensembles.

## 4.4 Application in Finance

In financial services, there are also several important applications. The original DYNOTEARS paper considered a model of diversification of investments in stocks based on dynamic Bayesian networks. Independently, (Ballester et al., 2023) consider systemic credit risk, which is one of the most important concerns within the financial system, using dynamic Bayesian networks. They found that the transport and manufacturing sectors transmit risk to many other sectors, while the energy sector and banking receive risk from most other sectors. To a lesser extent, there is a risk transmission present between approximately 25% of the sectors pairs, and these network relationships explain between 5 % and 40 % single systemic risks. Notice that these instances are much denser than the commonly used random ensembles.

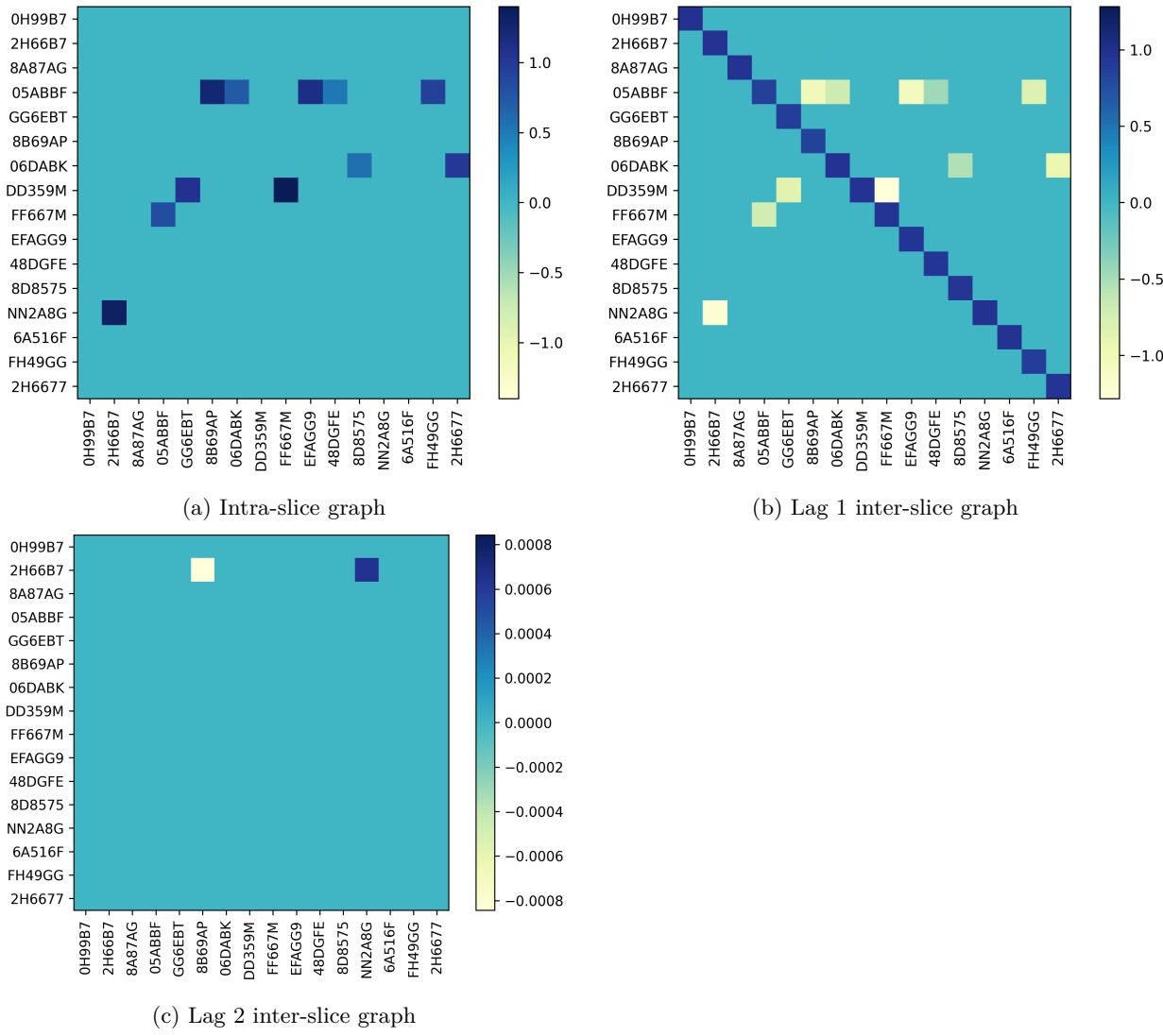

(a) Intra-slice graph

(b) Lag 1 inter-slice graph

(c) Lag 2 inter-slice graph

Figure 6: ExDBN - Heatmaps of the adjacency matrices of the learned intra-slice and inter-slice graphs for the CDS dataset.

We elaborate on the example of (Ballester et al., 2023), where 10 time series capture the spreads of 10 European credit default swaps (CDS). Considering the strict licensing terms of Refinitiv, the data from (Ballester et al., 2023) are not available from the authors, but we have downloaded 16 time-series capturing the spreads of 16 European CDS with RED6 codes 05ABBF, 06DABK, 0H99B7, 2H6677, 2H66B7, 48DGFE, 6A516F, 8A87AG, 8B69AP, 8D8575, DD359M, EFAGG9, FF667M, FH49GG, GG6EBT, NN2A8G, from

January 1st, 2007, to September 25th, 2024. The codes 48DGFE, 05ABBF, 8B69AP, 06DABK, EFAGG9, 2H6677, FH49GG, and 8D8575 belong to the Banks sector. The codes GG6EBT, DD359M, and FF667M belong to the insurance sector. Finally, the codes 0H99B7, 2H66B7, 8A87AG, NN2A8G, and 6A516F belong to transportation and manufacturing.

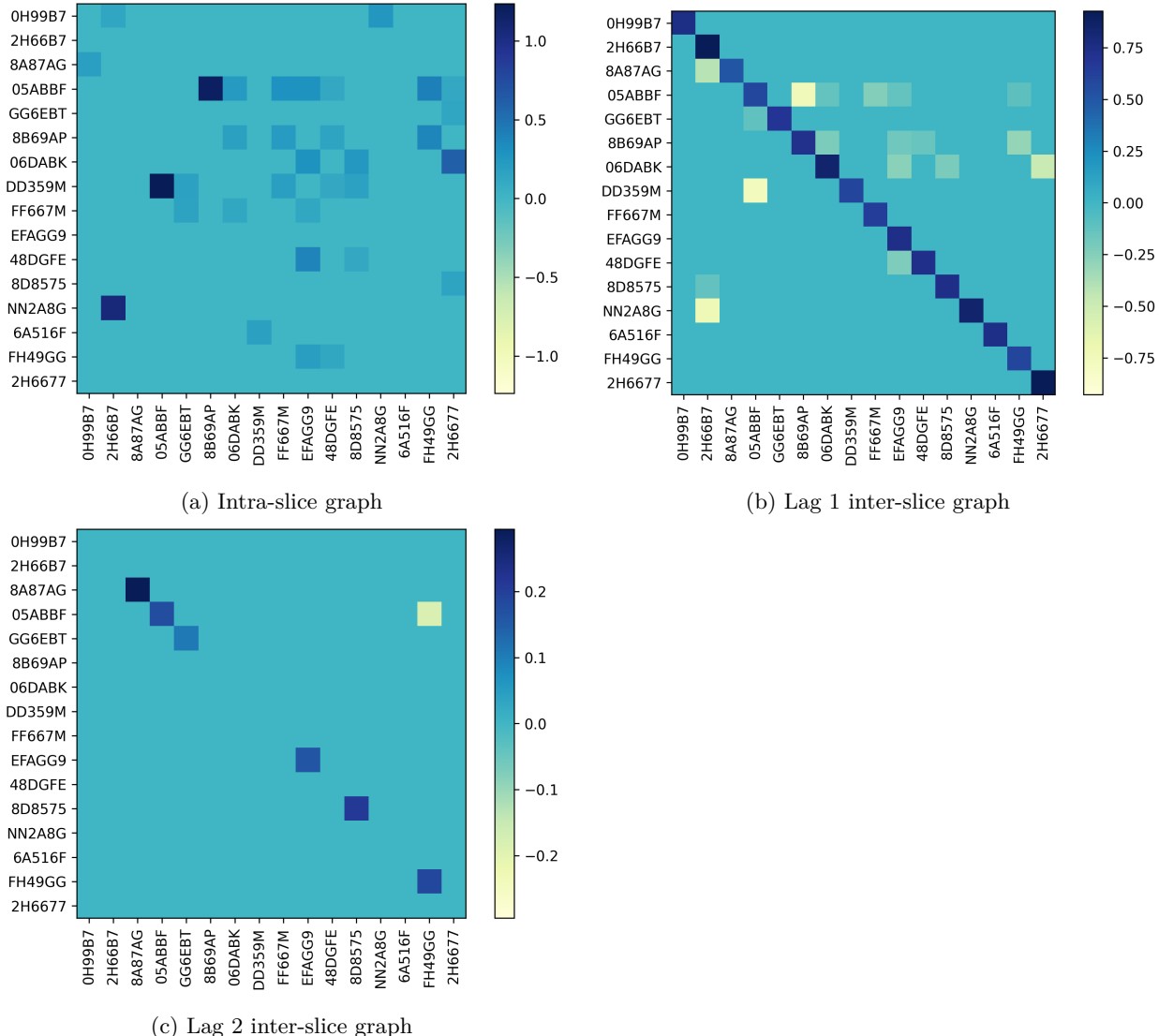

(a) Intra-slice graph      (b) Lag 1 inter-slice graph

(c) Lag 2 inter-slice graph

Figure 7: DYNOTEARS - Heatmaps of the adjacency matrices of the learned intra-slice and inter-slice graphs for the CDS dataset.

These data amount to more than 11 MB of time series data when stored as comma-delimited values in plain text. Although the procedure for learning the dynamic Bayesian network in (Ballester et al., 2023) is rather heuristic, we can solve the mixed integer programming (MIP) instance for the 16 European CDS in 30 minutes. In the heuristic of (Ballester et al., 2023), they first perform unconditional independence tests on each set of two time series containing an original series and a lagged time series, to reduce the subsequent number of unconditional independence tests performed. There are 45 unconditional and conditional independence tests performed first, to suggest another 200 conditional independence tests. We stress that the procedure of (Ballester et al., 2023) does not come with any guaranties, while our instance (11) is solved to a 20% MIP gap. The run-time of 30 minutes to a solution with a 20% MIP gap (using L2 regularization) validates the scalability of mixed-integer programming solvers.

For a solution of auto-regressive order 2, see Figures 6a, 6b, and 6c. We can see that each CDS mostly depends on its own history. Furthermore, as expected, the identified dependencies are mostly within sectors, with two exceptions: FF667M depends on 05ABBF and a lag 2 dependency is observed, with 2H66B7 depending on 8B69AP. For a comparison, we also included a solution found by DYNOTEARS; see Figures 7a, 7b, and 7c. In this solution, we can observe that the Lag 1 dependencies are very similar to those in the ExDBN solution. There are many additional Lag 2 self-dependencies, which we found rather redundant given that the same Lag 1 dependencies already exist. When examining intra-slice dependencies, we can identify one dependency between different sectors (DD359M–05ABBF). DYNOTEARS did not detect many strong intra-sector dependencies, unlike ExDBN. We would argue that the solution found by ExDBN is better, although this is somewhat subjective since the ground truth is unknown.

## 4.5 Application in Biomedical Sciences

In biomedical sciences, there is a keen interest in learning dynamic Bayesian networks to estimate causal effects (Tennant et al., 2020) and identify confounding variables that require conditioning. A recent meta-analysis (Tennant et al., 2020) of 234 articles on learning DAGs in biomedical sciences found that the averaged DAG had 12 nodes (range: 3–28) and 29 arcs (range: 3–99). Interestingly, none of the DAGs were as sparse as the commonly considered random ensembles; median saturation was 46%, meaning that each of all possible arcs appeared with a probability of 46% and did not converge to a global minimum of the problem.

As an example, we consider a recently proposed benchmark of (Ryšavý et al., 2024), where the Krebs cycle is to be reconstructed from time series of reactant concentrations of varying lengths. There, DYNOTEARS cannot reach the (Ryšavý et al., 2024) F1 score of 0.5 even with a very long time series. In contrast, our method can solve instances (4) to global optimality. Using ExDBN, however, the global minimization is ensured given sufficient time and thus the maximum likelihood estimator is found. However, it should be noted that depending on the number of samples and noise, it may be that even the maximum-likelihood estimator is not sufficiently accurate. However, this does not reflect poorly on the method itself, but it is rather a matter of the modification of data collection methods associated with the experiment. In a one-hour time limit, ExDBN can find a solution with the 38% duality gap.

## 5 Conclusion

Dynamic Bayesian networks have wide-ranging applications, including those in biomedical sciences and computational finance, as illustrated above. Unfortunately, their use has been somewhat limited by the lack of well-performing methods to learn them. Our method, ExDBN, provides the best possible estimate of the DBN, in the sense of minimizing empirical risk (3). Significantly, our method does not suffer much from the curse of dimensionality, even for real-world dense instances, which are typically challenging for other solvers. This is demonstrated most clearly in the case of systemic risk transmission, detailed in Section 4.4, where a solution with 20% MIP gap is found in 30 minutes. Additionally, the use of the guarantees on the distance to the global minimum (so-called MIP gap, available ahead of the convergence to the global minimum) provides a significant tool for fine-tuning the parameters of the solver in the case of real-world application, where the ground truth is not available. Combined with global convergence guarantees of the maximum likelihood estimator, this provides a robust method with state-of-the-art performance.

**Acknowledgments**

The authors acknowledge the support of National Recovery Plan funded project MPO 60273/24/21300/21000 CEDMO 2.0 NPO. This work has been partially supported by project MIS 5154714 of the National Recovery and Resilience Plan Greece 2.0 funded by the European Union under the NextGenerationEU Program.

**Disclaimer**

This paper was prepared for information purposes and is not a product of HSBC Bank Plc. or its affiliates. Neither HSBC Bank Plc. nor any of its affiliates make any explicit or implied representation or warranty

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
