# OpenReview forum: "ExDBN: Learning Dynamic Bayesian Networks using Extended Mixed-Integer Programming Formulations"
_TMLR — Accepted by TMLR_

### Review · Reviewer_ghVQ · 2025-08-14

**Summary Of Contributions:**

The paper proposes a score-based algorithm for learning directed acyclic graphs with time lags using a mixed-integer quadratic programming formulation. By employing a Branch-and-Bound-and-Cut strategy, the algorithm effectively mitigates the curse of dimensionality by avoiding the need to regenerate exponentially many acyclicity constraints.

Empirical results on synthetic data show that the proposed method outperforms DYNOTEARS in certain scenarios, particularly with small sample sizes and a moderate number of nodes, while remaining competitive with DYNOTEARS on larger datasets.  The algorithm has also been applied to two real-world datasets.

**Audience:**

Yes

**Broader Impact Concerns:**

No concerns.

**Claims And Evidence:**

No

**Requested Changes:**

1. It would be helpful to provide a clearer explanation of the Branch-and-Bound-and-Cut approach, similar to the level of detail given in other parts of the methodology section.

2. Including additional baselines would strengthen the empirical evaluation. While the practical limitation of benchmarking against all existing solvers is understandable, the use of small sample sizes and graph scales may help mitigate this constraint, allowing for the inclusion of multiple representative baselines. Additionally, there are some baseline methods in the same line of work as DYNOTEARS that are not computationally intensive. Please consider reviewing and incorporating such methods as additional baselines.

3. Would it be feasible to report the number of constraints generated or applied by the Branch-and-Bound-and-Cut approach? Doing so could provide a more concrete illustration of how the proposed algorithm addresses the curse of dimensionality.

4. For real-world datasets, please include comparisons using the same evaluation metric if ground truth is available. If not, please clarify whether the estimated DAG is consistent with domain knowledge and explain its practical implications. Figure 4 is not informative without contextual analysis.

5. Please clarify why the proposed algorithm is only applied to graphs with up to 25 nodes. For reference, DYNOTEARS has been applied to graphs with up to 100 nodes.

**Strengths And Weaknesses:**

**Strengths**

1. The proposed algorithm addresses the curse of dimensionality, which is a significant and relevant challenge both theoretically and practically.
2. The algorithm has been evaluated through a comprehensive set of experiments, including both synthetic and real-world datasets.
3. The method appears to be consistent.

**Weaknesses**

1. Section 3, which covers the methodology, should be a core part of the paper but currently lacks sufficient explanation and detail.
2. Although the algorithm is intended to tackle the curse of dimensionality, it is only applied to small- and medium-sized causal graphs (up to 25 nodes), raising concerns about its scalability.
3. The experimental evaluation is limited, as only DYNOTEARS is used as a baseline. In the biomedical application, comparing the F1 score of DYNOTEARS with the duality gap of the proposed method is not straightforward. Similarly, in the finance application, there is neither a clear performance comparison metric, nor an explanation of the real-world implications if the ground truth is unavailable.

---

> ### Author Response · Authors · 2025-09-21
>
> Dear Reviewer,
> Thank you for your review. Let us address your concerns and questions.
>
> 1) We added a short paragraph explaining the Branch-and-Bound-and-Cut algorithm in the revised version.
>
> 2) We are not aware of any recent methods that outperform Dynotears and Lingam. If you could point us to such methods, we would be glad to run additional benchmarks, provided source code is available.
>
> 3) We added a graph showing the number of lazy constraints added in each scenario.
>
> 4) The ground truth is not available due to the nature of the problem. We added more description of the codes, which correspond to CDS in the various sectors, as well as a short discussion of the solution found by ExDBN in the revised version of the article.
>
> 5) We limited benchmarks to 25 nodes due to computation time and memory constraints. To generate Figures 1 and 2, we had to run ExDBN approximately 15,000 times, with a two-hour limit per run. This required about one week of computation on our cluster within our quota.

---

> > ### Comment · Reviewer_ghVQ · 2025-09-26
> >
> > Dear author, thank you for the response.
> >
> > For item 2, I think NTS-NOTEARS has outperformed DYNOTEARS on two benchmark datasets.
> >
> > For item 5, is there any discussion about the scalability of the proposed algorithm? I thought tackling the curse of high dimensionality was one of the main contributions of the proposed algorithm, but in practice, one big challenge is still the running time on high-dimensional data? Also, it is fine not to perform experiments from small $n$ to large $n$; increasing dimensionality solely with a fixed $n$ is fine to temporarily show the experimental results so you don't need to run ExDBN so many times.

---

> > > ### Author Response · Authors · 2025-10-13
> > >
> > > Dear reviewer, thank you for your questions. We address them below:
> > > 1. We included NTS-NOTEARS in our benchmarks. In our settings, ExDBN outperforms NTS-NOTEARS. We acknowledge that prior results show NTS-NOTEARS outperforming DYNOTEARS on two benchmark datasets. We believe the discrepancy stems from NTS-NOTEARS being optimized for non-linear SEMs, whereas our benchmarks emphasize the linear structural equation models.
> > >
> > > 5. We ran additional experiments up to 80 variables (see Figure 5). For this number of variables, ExDBN still outperforms the other methods on our benchmark. We could not run ExDBN beyond 80 variables because the Gurobi solver exceeded our available memory (128 GB), which limits the scalability of the method. Therefore, ExDBN is particularly suitable when high-quality solutions are needed for medium-sized instances. We also updated the abstract to note that comparisons were performed on up to 80 variables.

---

### Review · Reviewer_Tk3c · 2025-08-20

**Summary Of Contributions:**

The paper proposes a score-based framework—ExDBN—for learning Dynamic Bayesian Networks by jointly estimating intra-slice DAGs and inter-slice (lagged) dependencies via a mixed-integer quadratic program with sparsity or shrinkage regularization, enforcing acyclicity only within slices and certifying solution quality through the MIP gap. Algorithmically, it advances branch-and-cut for dynamic graphs by adding lazy cycle constraints on demand (with an empirical recommendation to add all detected cycles per iteration), which avoids exponential pre-enumeration while retaining global optimality. Empirically, across synthetic ER and scale-free benchmarks and multiple AR orders, ExDBN achieves lower SHD and higher F1 than strong baselines such as DYNOTEARS and LiNGAM, particularly in small-sample or heteroskedastic settings and at larger graph sizes. Two case studies—a biochemical Krebs-cycle network and a systemic-risk application using European CDS spreads—demonstrate practical viability on real data, including instances solved to global optimality.

**Audience:**

Yes

**Claims And Evidence:**

Yes

**Requested Changes:**

**Professionalism & presentation issues (requested fixes).**

1. Abstract format. Make the abstract a single, self-contained paragraph.
2. References. Several entries appear manually typed and contain errors; please generate the bibliography from a .bib file to ensure consistent author names, titles, venues, capitalization, and dates. For example, the following contain obvious issues and should be corrected using canonical metadata:
– Tian Gao, Debarun Bhattacharjya, Elliot Nelson, Miaoyuan Liu, and Yue Yu. “Idyno: Learnmalinskying nonparametric dags from interventional dynamic data.” In ICML, 2022.
– Jakob Runge. “Discovering contemporaneous and lagged causal relations in autocorrelated nonlinear time series datasets,” 03/2020.

3. Citations on first mention. Add a citation immediately after the first mention of each solver/algorithm (e.g., SCIP, GLPK, Gurobi, CPLEX) and for standard procedures like DFS.
4. Equation referencing style. Refer to numbered displays with parentheses: write Equation (7) or Inequality (7), not “Equation 7.” Use a single, consistent style throughout.
5. Equation typesetting. Avoid splitting equations into multiple lines with repeated “==” or “++”.
6. Redundant dating in prose. Replace formulations like “In 2020, Pamfil et al. (2020) have developed…” with a concise form, e.g., “Pamfil et al. (2020) developed…” or “DYNOTEARS (Pamfil et al., 2020) is…”. Avoid leading with the year and do not repeat it in parentheses.

**Strengths And Weaknesses:**

**Strengths**

1. The paper propose a mixed integer progamming and avoid exponentially many acyclicity constraints by utilizing the so"lazy constraint".

2. The paper includes real‐data case studies and reports improvements in both runtime and structural accuracy over baselines.

**Weakness**

1. There are numerous presentation/professionalism issues (e.g., typos, imprecise notation); see Requested Changes for a detailed list.

2. Constraint (14) is unclear: as written, it seems to require selecting $|C|-1$ edges from a DAG $C$, which is unintuitive and, to me, likely incorrect—please clarify the intention and provide a justification or correction.

3. The motivation for focusing specifically on Dynamic Bayesian Networks is not well articulated. The proposed MIQP/constraint trick appears applicable to static Bayesian network (DAG) structure learning as well; please clarify what is genuinely unique to the dynamic setting (or provide static-BN results to delineate the contribution).

---

> ### Author Response · Authors · 2025-09-21
>
> Dear Reviewer,
> Thank you for your review. Let us address your concerns and questions.
>
> 1) We implemented all the requested changes in the revised document.
>
> 2) There is one constraint for each possible directed cycle in a complete directed graph on $n$ vertices. For each such cycle $C$, the edges in the sum belong to the directed cycle $C$, which has length $|C|$. No edge is selected from any DAG. For example, if $C$ forms a triangle with three vertices {1,2,3} and three edges {a,b,c}: $1 \xrightarrow{a} 2 \xrightarrow{b} 3 \xrightarrow{c} 1$, then the sum is over the edges ${a,b,c}$.
>
> 3) Indeed, if we set the autoregression order to 0, the method could be applied to static Bayesian networks. However, we believe that static and dynamic Bayesian networks are quite different topics. There are far more solvers available for static Bayesian networks than for dynamic ones. If we would add the static case to the article, most benchmarks in the article would concern static Bayesian networks, which was not the original focus of this work. Thus, the uniqueness here lies in the availability of solvers for the dynamic case.

---

### Review · Reviewer_vDaK · 2025-08-25

**Summary Of Contributions:**

This work proposes a new mixed-integer quadratic program formulation for causal learning from data. To capture dynamic effects, the authors assume a Dynamic Bayesian network setting, where a structural equation model contains both intra-dependencies to the present as well as inter-dependencies on past data. The standard way of solving such a problem would incur exponentially many acyclicity constraints, making the method impractical. Instead, the authors avoid pre-generating all these constraints by lazily adding constraints in a branch-and-cut manner. The authors compare their approach to DYNOTEARS and lingam under various settings, and show that the proposed method produces more accurate results with small- and medium-sized synthetic instances containing up to 25 time series. Furthermore, they demonstrate the benefits of their method, particularly in terms of global optimum convergence, in two applications in bioscience and finance.

**Audience:**

Yes

**Broader Impact Concerns:**

I do not have broader impact concerns.

**Claims And Evidence:**

No

**Requested Changes:**

The proposed method is interesting, but I feel the paper would benefit from various revisions:
- The authors could explain why the third variant of lazily adding constraints is better by conducting an ablation study. Currently, it looks as if they did experiments, but do not provide any details at all. Furthermore, it seems that their finding may contradict earlier findings, e.g., (Achterberg 2007, Chapter 8.9). If so, more details are necessary.
- I'm not so sure that having only two competitors from 2010 and 2020 is sufficient proof about the superiority of the proposed method. The authors should either include more recent methods, or at least explain why they have not included more recent methods (e.g., due to scope, restrictions of more recent methods, etc.).
- Why are Figures 1 and 2 not better aligned? As I argued above, the results on the two metrics are not totally consistent across all datasets. For example, in Figure 2 it is not the case that as the number of nodes in the DAG increases, the performance gap between ExDBN and DYNOTEARS increases, too (for n=20 or n=100). Furthermore, Figure 2 does not convincingly show that ExDBN is better than DYNOTEARS. The authors could elaborate on all these aspects in the revised manuscript. The results do not necessarily look very convincing, especially in Figure 2, which is further exacerbated by the fact that the authors have only included two old competitors.
- ExDBN can often suffer from significant variance. Why is that the case? Can the authors elaborate on that, and argue why this is not a problem in practice?
- I was not really clear about the purpose of the finance experiment. The authors do not provide any result on the solution quality of other methods. Is the point to simply argue that ExDBN can converge to the global optimum?

**Strengths And Weaknesses:**

Strengths
- The authors study a very important problem. The fact that the setting captures time dependencies has high practical potential in important real-world applications.
- The mixed integer programming formulation is interesting, because it allows the use of globally convergent algorithms. As opposed to formulations in the prior literature, the setting under study involves a directed intra-dependency graph, which makes the problem more challenging.
- The idea the authors propose of lazily adding constraints during branch-and-cut is interesting, as it enables a practical globally convergent algorithm (even though the actual convergence speed may be slow, depending on the setting).
- The authors include synthetic experiments where the ground truth is known, so that the performance metrics can be accurately tracked and analyzed. The results seem to suggest that the proposed framework can have a positive impact for small- and medium-sized datasets (e..g, n=50 or n=100), especially as the number of nodes increases. On the other hand, for n>=250 the benefit seems to be negligible or non-existent. Furthermore, in the biomedical example, ExDBN can in principle find the global optimum, as opposed to DYNOTEARS that cannot even reach an F1 score of 0.5.

Weaknesses
- The authors do not explain why the third variant of lazily adding constraints is better than the other two, and do not provide any results at all.
- I have concerns with regard to the competitors. The authors compare against DYNOTEARS, a method from 2020, and lingam, a method from 2010. I understand the argument that they do not need to include methods that DYNOTEARS outperforms. But what about more recent methods? The authors have cited more recent works in the field, but for some reason they did not include any comparison. I feel that this point needs clarifications.
- Figure 1 seems to indeed suggest that ExDBN outperforms the studied competitors for small- and medium-sized synthetic datasets, especially as the number of nodes increases. On the other hand, for n=250 and n=500, we see that ExDBN can occasionally perform relatively poorly (e.g., in SF-3-1 and ER-2-2). But the biggest concern was probably with Figure 2, since the F1 score does not closely track the SHD score and the results are not fully consistent. For instance, for n=50 and ER-2-2, ExDBN is generally outperformed by DYNOTEARS in Figure 2, whereas the opposite is True in Figure 1. The two scores seems not to be so well aligned. And in Figure 2 the performance of ExDBN is not always better (look e.g., at AR-2-2 across all values of n).
- ExDBN seems to suffer from high variance, compared to both lingam and DYNOTEARS. In some cases, the variance is especially pronounced, e.g., ER-2-2 for n=50.
- The bioscience example provided an data point (DYNOTEARS cannot even reach an F1 score of 0.5, as opposed to ExDBN that can find a solution with 38% duality gap). I'm not sure there was such a comparison for the finance dataset. For instance, what solution did the procedure of Ballester et al. (2023) converge to?

---

> ### Author Response · Authors · 2025-09-21
>
> Dear Reviewer,
> Thank you for your time and your review. Let us address your questions and concerns.
>
> 1) We ran an experiment on the SF3-1 scenario with three variants of adding lazy constraints, and obtained the following results. On average, the MIP gap is 0.335 when adding constraints for all found cycles, 0.344 when using only the shortest cycle, and 0.346 when adding the constraint corresponding to the first found cycle. We have added these results to the revised version of the paper.
>
> 2) We have not found any more recent methods that perform better than Lingam and Dynotears. If you could point us to such a method, we would gladly run additional experiments for comparison, provided that source code is available.
>
> 3) You are correct. By accident, the data in Figure 2 corresponded to different hyperparameter settings than in Figure 1. We have regenerated Figures 1 and 2. Please see the revised version of the document.
>
> 4) The large variance of ExDBN in our benchmark is due to the uniform two-hour computation time limit. Some instances require more time to achieve a good solution. Of course, high variance can be problematic in practice. In a real application, one could monitor the MIP gap and stop the computation once the gap is sufficiently small. However, this approach is somewhat tricky, as there is no guarantee it will perform well on every application. We therefore believe the method should be considered as one of several approaches in a portfolio, rather than as a standalone solution.
>
> 5) We aimed to provide an example application on real-world problems. Since the ground-truth graph is unknown, it is difficult to evaluate solution quality in the same way as with synthetic data. The only guarantee we can provide is convergence to the global optimum. We have added further discussion of the solution found by ExDBN to the revised version of the article.

---

> > ### Comment · Reviewer_vDaK · 2025-09-23
> > **questions**
> >
> > Thank you for the response. Some questions:
> > - The gain compared to DYNOTEARS is mostly taking place for smaller datasets, up to 100 samples. Above 100 samples, the new framework seems to perform almost identically to DYNOTEARS. Is this understanding accurate, and, if yes, do you have some explanation for that? What makes small datasets particularly challenging for DYNOTEARS (compared to the proposed framework)?
> > - Can you elaborate more on the execution time of their framework? Is it comparable to DYNOTEARS? Is it more demanding? The paper acknowledges that the cycle detection component can negatively affect the running time; furthermore, the new Figure shows that dozens of thousands of constraints can be added during runtime. The aggregate effect on performance though is rather unclear.  Solution quality is probably the most important metric, but runtime is a significant aspect as well, and I feel it is nor really covered in the current draft.
> > - For the application in finance, do you have results from DYNOTEARS to report? A performance comparison between the two frameworks in this application domain would be interesting.

---

> > > ### Author Response · Authors · 2025-10-13
> > >
> > > Thank you for your comment. Let us address your questions:
> > > 1. We believe that DYNOTEARS tends to end up in a local minimum, which ExDBN can avoid, especially for smaller datasets, where it can reach the global minimum within the given time limit.
> > > 2. ExDBN is more demanding than DYNOTEARS in both time and memory. We set the time limit for ExDBN to two hours, while DYNOTEARS usually converged within a few minutes. ExDBN is more suitable in scenarios where obtaining a high-quality solution is more important than computing it quickly. We have also added information about memory usage to the article.
> > > 3. We added the solution found by DYNOTEARS and a discussion that explains the differences between the ExDBN and DYNOTEARS solutions, as well as why we believe ExDBN performs better, in Section 4.5. We also clarified that in 30 minutes ExDBN finds a solution with a 20% MIP gap rather than the global optimum. Note that 20% represents an upper bound on the solution quality.
> > >
> > > Finally, we added additional experiments comparing NTS NOTEARS and a simplified benchmark that evaluates the methods up to 80 variables.

---

### Decision · Action_Editor_Hw2N · 2025-10-21

**Recommendation:** Accept with minor revision

**Additional Comments:**

All reviewers found issues in the original submission that warranted a revision of the manuscript, ranging from limited baselines in empirical results, lacking ablations, mistakes in figures, and poor writing conventions. Since then, a substantial revision has been made addressing most of the reviewers' concerns. Ultimately, the authors have written responses to the reviewers' questions and requested changes, some of which have been used to revise the paper further after the last interaction with reviewers.

I'm recommending that the paper go through a final minor revision to incorporate the remaining suggestions by reviewers, some of which were discussed in questions and their responses:
* A discussion comparing the running time with baselines and the limitations of scalability in the present algorithm
* A discussion of how the execution time limit interacts with the variance of the estimates
* A comparison with one of the baselines on the finance data set
* A more in-depth discussion of the differences between the static and dynamic BN problems to justify the present study

**Audience:**

Yes

**Audience Explanation:**

Several of the reviewers noted the importance of the problem.

**Claims And Evidence:**

Yes

**Claims Explanation:**

The authors make modest claims of empirical advantages over baselines on synthetic instances. These empirical results are presented clearly enough.

---

> ### Author Response · Authors · 2025-11-12
>
> Dear Editor,
>
> Thank you for your time and comments. We have submitted the camera-ready version of our article and have addressed your suggestions as follows:
> 1. We added Table 1 with the average running times of each method and modified the discussion of scalability in the second-to-last paragraph of Section 4.3.
> 2. We added a note above Figure 4 regarding the ExDBN variance and performed additional experiments, added to Figure 3, comparing standard deviation and running time. In the previous revision, Figure 3 contained only one run of ExDBN; it now shows an average over ten runs with the standard deviation displayed as a grey area.
> 3. We already included a comparison with DYNOTEARS in the previous revision. Could you please clarify if you meant something else?
> 4. We added a paragraph discussing the differences between BNs and DBNs before the beginning of Section 2.1.

---

> > ### Comment · Action_Editor_Hw2N · 2025-11-19
> > **Re:**
> >
> > Looks good! Thank you.
> >
> > Could you also insert the year/month that shows in the heading, as instructed in the style file?

---

> > > ### Author Response · Authors · 2025-11-19
> > >
> > > We added the year and month. Thank you.